# Intersection and Flattening of Surfaces in 3D Models through Computer-Extended Descriptive Geometry (CEDG)

Manuel Prado-Velasco * and Laura García-Ruesgas

Department of Engineering Graphics, University of Seville, 41092 Seville, Spain; lauragr@us.es
* Correspondence: mpradov@us.es

**Abstract:** Computer-extended Descriptive Geometry (CEDG) is a new approach to solving and building computer models of three-dimensional (3D) geometrical systems through descriptive geometry procedures (thus inheriting invariant-symmetry properties from projective geometry) that have demonstrated reliability and accuracy. CEDG may calculate a parametric implicit functional form for the spatial curves generated in the intersection of two surfaces, as well as of the flattened pattern of any developable surface involved in those encounters. This study first presents the theoretical foundations and methodology to calculate those curves. Secondly, a compound hopper is defined and modeled through CEDG (implemented in GeoGebra) and CAD (Solid Edge© 2023) approaches to evaluate the advantages of CEDG against CAD. The results demonstrate the robustness and accuracy of the CEDG technique for he intersection and flattening of surfaces and the advantages of CEDG against Solid Edge 2023 in solving the hopper case study.

**Keywords:** descriptive geometry; computer graphic parametric modeling; CAD; CEDG; dynamic geometry software

## 1. Introduction

Descriptive geometry (DG), which inherits the invariant-symmetry properties of projective geometry, is a science that was systematized by Gaspard Monge in 1794, and its associated representation systems, such as the dihedral system, have determined the foundations of graphic representation in engineering since the second half of the 19th century. This discipline has huge importance in the development of spatial vision and constitutes a fundamental pillar in university teaching, although its use in the professional field has been practically replaced by computer-aided design (CAD) software [1]. CAD technology emerged in the mid-1990s as a computer-based approach to create, modify, analyze, and document 2D or 3D graphical representations of physical elements, providing an alternative to manual drafts and product prototypes. The initial main reason for the replacement of DG by CAD systems lies in the current wide availability of software packages capable of representing three-dimensional shapes. The geometry of the CAD model is essentially represented by curves and approximation surfaces, such as B-splines, which provide high control and accuracy [2]. The subsequent evolution of CAD gives tools to model very complex surfaces through Non-Uniform Rational B-Splines (NURBSs) such as, for instance, those that define a human vascular system [3], overcoming the scope of DG.

The determination of the intersection between two surfaces has been a problem in CAD for around 60 years and continues to be an active topic of research today. The main reason lies in the fact that the technique used has to balance three conflicting objectives, accuracy, strength and efficiency [4]. The problem of intersection between surfaces involves the design of algorithms suitable for evaluating and performing geometric operations, as well as their representation. The exact requirements of the representation and algorithms depend on the particular application. Generally speaking, any representation should

provide functionality to perform the operations of evaluating and rendering the intersection curve, deciding whether a point belongs to the intersection curve, organizing the points that lie on the intersection curve, and using it as the boundary edge of a trimmed surface.

The evaluation of the intersection between two rational parametric surfaces is a recurring operation in solid modeling and can be approached using algebraic sets in a higher dimensional space. Using results from elimination theory, it is possible to project the algebraic set onto a lower dimensional space, using the matrix itself as the representation of the algebraic set in this space. Given this representation, properties of rectilinear programs and linear algebra results are used to perform geometric operations on the intersection curve. The accuracy of these operations can be improved by pivoting or other numerical techniques [5].

Nearly 30 years ago, the strength deficiencies shown in the methods of intersection between surfaces applied at the time [6] were being addressed. For this purpose, exact methods based on rational arithmetic (oriented to rational B-spline surfaces) began to be used, techniques that have been transferred to the usual CAD.

Subsequent studies reviewed the problem of intersection between surfaces for CAD/CAM, also developing a new algorithm based on the geometric properties of the surfaces and their intersection curves, establishing an integration of the segmentation and tracing methods [7]. This algorithm translates the nonlinear problem of finding the surface–surface intersection (SSI) curve into a series of steps for solving sets of linear equations. The starting points for the plotting are computed using a combination of numerical and subdivision techniques. In the plotting phase, the intersection curve is drawn in the direction of the combination of its tangent vector and the direction of the geodesic radius of curvature, and the size of the steps is adaptively adjusted according to the curvature of the intersection curve. Nevertheless, at the singular points, only the tangent direction was taken into account. As an evolution of the previous methods, in [8], two hybrid models were generated for the calculation of the intersection of two Bezier surfaces, obtaining results that are more accurate and therefore closer to the real solution than using the plane/plane intersection. Two matching methods were used to calculate exactly the intersection points, employing tangential or circular steps. An important difference between the two techniques lies in the step size. In the method using tangential steps, the step size must be fixed and predefined, while in the circular step method, the step size is dynamic and changes automatically. The choice of techniques to use will depend on the particular problem. If the result is twisted or sinuous, the circular pitch is recommended; otherwise, it is enough to use the tangential pitch.

A new improvement to the previous surface intersection methods was presented in [9] for surfaces based on NURBS. The results obtained in this study allow some assumptions to be eliminated that were necessary to apply in the parameterization of surfaces, thus facilitating this process for two general surfaces. Finally, recent studies present new techniques targeting toroidal patches in which the intersection curve of two free-form surfaces is computed by employing a bounding volume hierarchy (BVH), where the leaf nodes contain oscillating toroidal patches. The effectiveness of the technique applies, among other cases, to two nearly identical surfaces ([10], Figure 7).

Computer-Extended Descriptive Geometry (CEDG) constitutes a new way of undertaking computer modeling of 3D geometric systems [11], trying to offer a solution to the previously identified constraints of the current CAD systems [6]. As discussed, CAD tools favor the construction of virtual prototypes of 3D systems that can be manipulated in space and projected in a simple way according to the chosen representation system. Nevertheless, they do not allow the model to be generated when it depends on some implicit parameter (which can only be measured when the model is built). In addition, they still show important deficiencies in the calculation of flat patterns of sheet metal surfaces.

The CEDG approach combines the ability of DG to solve spatial geometric problems with the skill of dynamic geometry [12] in the building of geometric-mathematical models. A CEDG model is a flat sequence of mathematical entities that may be associated with

graphical objects in a one-to-one relationship. The entities consist first of 2D primitive objects (point and lines), and other 2D curves, which combine themselves to produce projections of 3D objects according to DG. Any entity in the CEDG model keeps mathematical dependencies with other previously built entities. In this manner, any change in an entity's parameter propagates to the dependent entities in a dynamic manner, ensuring the consistency of the model. The model is completed with a second collection of mathematical entities that implements mathematical procedures from DG and physical relationships that could be required (e.g., forces equilibrium applied to a system).

The CEDG modeling is implemented in a dynamic geometry system working under the deterministic condition (in opposition to continuous), to extend the mathematical procedures of DG with the aim of describing the projections of 3D curves through implicit parametric functions. To the best of our knowledge, this is the first approach that uses the DG procedures to build parametric computer 3D models in a manner that exploits the body of knowledge of DG, which includes the preservation of the integrity of curves and surfaces, in contrast to CAD systems. The CEDG approach gives a response to the need for DG in the world of 3D modeling [13], which is opposed to the opinion that DG is dead Ortiz-Marín et al. [14].

The underlying concepts of CEDG appeared in the PhD thesis of Wottreng [15], although the research was limited to the analysis of the influence of the developments of Gaspard Monge on descriptive and differential geometry. A preliminary version of the CEDG approach implemented on the dynamic geometry software Geogebra [16] demonstrated its reliability to model some 3D systems [11].

The objective of this study is the development of theoretical methods for the computation of encounters between surfaces and for their flattening based on CEDG. For this purpose, a methodology that is different from those presented so far will be used.

The usual procedure used in DG to calculate the intersection between two surfaces consists of defining it by means of their projections, which are obtained through interpolation on a set of points belonging to them. The points are obtained by means of DG techniques, which are applied iteratively so that the accuracy and complexity of the resulting curve are proportional to the number of points attained [17]. The CEDG approach extends the DG procedures to find a parametric mathematical function that defines the sought intersection curve. The parametric function is extracted by means of a locus function from the geometrical constructions of the model, which is computed from the sequence of geometric-mathematical entities associated with the calculation of a single generic point of the intersection [11].

This paper develops a theoretical technique that computes the intersections and flattening (first objective), which is used to solve the case study defined by a compound hopper, whose CEDG result (3D model and flat pattern) is compared with the solution obtained using Solid Edge 2023 (second objective). The details of the comparison are presented in the Section 2.

## 2. Materials and Methods

The study has two stages. The theoretical development of the CEDG technique for surfaces' intersection and flattening is presented in the *first stage*. We use the parametric 3D model of an oblique cone that intersects with an oblique cylinder (Figure 1) and is subsequently flattened, to clarify the meaning and scope of the mathematical entities and procedures involved in the novel method.

The variables and other terms used in the theoretical development are defined at their first use. Notwithstanding, those that refer to especially relevant objects are listed also in the Abbreviations section. The 3D surfaces and entities are defined by their vertical and horizontal projections.

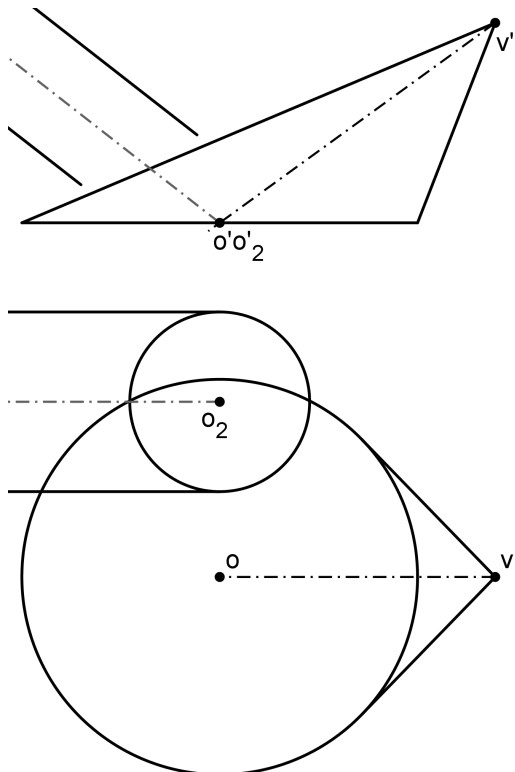

**Figure 1.** Cone–cylinder intersection for illustrating the theoretical development (first stage).

The technique is applied in the *second stage* to solve the hopper defined by the projections of Figure 2, computing the surface intersections and flat patterns (flattening) of the upper duct, $D_u$ ($d'_u - d_u$), and the lower duct, $D_l$ ($d'_l - d_l$), and comparing them against the flat patterns obtained through Solid Edge 2023. According to the methodology pointed out in Prado-Velasco and Ortiz-Marín [17] the flat pattern refers to the neutral surface (fiber) of the hopper.

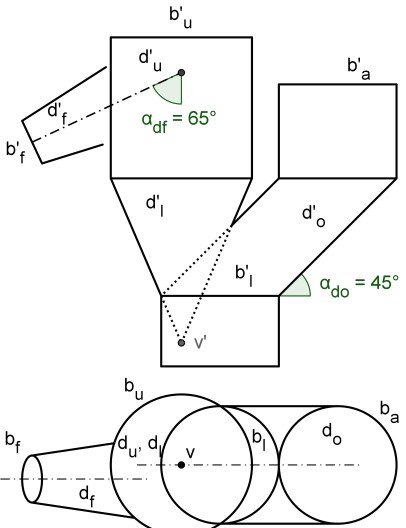

**Figure 2.** Hopper defined through its projections (second stage).

The hopper is completed by a fluid duct, $D_f$ ($d'_f - d_f$), which transports fluid towards the upper duct, and an oblique duct, $D_o$ ($d'_o - d_o$), where aggregates (small discrete mass of coarse to medium particles) from the aggregates border, $B_a$ ($b'_a - b_a$) go into the system. The fluid duct is a conical surface that discharges into $D_u$ through the discharge area, $A_d$,

which, assuming that the system operates in a steady state, may be defined through the following Equation (1), in which $G$ is the mass flow rate of fluid through $D_f$ and $\rho$, $\vec{v}$ and $\vec{dS}$ are the density and velocity of the fluid and a differential element of area perpendicular to $\vec{v}$ at the position $r$ in the discharge surface, $S_d$. Thus, $\rho_d$, $v_d$ and $A_d$ are average values of density and fluid velocity and the area of the discharge section.

$$G = \int_{S_d} \rho(r)\vec{v}(r)\vec{dS}(r) = \rho_d v_d A_d \, , \tag{1}$$

The model needs to be optimized to achieve a very fast expansion of the fluid in the last portion of the conical duct (quasi—adiabatic expansion). With that aim, the 3D model must facilitate the research on the dimensional parameters that provide a fluid expansion given by the outlet/inlet area ratio $A_r = A_d/A_i > 3$ ($A_i$ is the fluid duct inlet area). The value of $A_r$ is controlled by the $D_u - D_f$ connection curve, which depends on the conicity of $D_f$ (Con) and the horizontal distance between the $D_f$ axis and the horizontal projection of vertex, v (Eccentricity), Ecc, whereas the remaining dimensions of the hopper are constant. The conicity induces a smooth increment of area, whereas the eccentricity promotes a greater surface inside the $D_u - D_f$ connection curve without a gradual increment along the fluid duct.

The other hopper dimensions taken as parameters are the diameter of the inlet fluid border ($b'_f - b_f$), $D_{df}$, the diameter of the upper border ($b'_u - b_u$), $D_{bu}$, the diameter of the lower border ($b'_l - b_l$), $D_{bl}$, the height of the lower duct with respect to its vertex, $H_{ld}$, the angle of fluid duct axis with respect to the vertical, $\alpha_{df}$, and the angle of the oblique duct with respect to the horizontal, $\alpha_{do}$.

The discharge surface, $S_d$, may be defined by the cylindric surface inside the warped intersection curve between surfaces $D_f$ and $D_u$, which will be computed during the hopper modeling. According to the Equation (1), the area $A_d$ may be considered perpendicular to the streamline. We are interested in performing an analysis focused on the 3D model and independent of the fluid system. Therefore, we will use the 3D hopper model to propose and compute a surrogate of $A_d$.

We define the following three specific targets in this second stage:

1. Feasibility to reach the parametric 3D hopper's model and the required flat patterns.
2. Application of the 3D model to achieve $A_r > 3$ with minimum conicity through the eccentricity dimension.
3. Accuracy of the $D_u$ and $D_l$ flat patterns.

The values of model parameters were grouped in two main dimensional groups (Nom and Var) and several variations within these ones, as shown in Table 1. Each row represents the values of the dimensions referred to (Con, Ecc, $D_{df}$, $\alpha_{df}$ and $\alpha_{do}$) that were assigned to evaluate the models according to the above specific targets.

The parameters that are not set in this table were kept constant during the study. These are $D_{bu} = 6$ m, $D_{bl} = 5$ m, $H_{ld} = 7$ m, $\alpha_{df} = 65°$, and $\alpha_{do} = 45°$. Nonetheless, they were modified to verify that the final model responds properly to their change.

**Table 1.** Dimensions' groups values (meters) for 3D model CEDG–CAD comparison.

| Dim. Group | Con [†] | Ecc | $D_{df}$ | $\alpha_{df}$ [‡] | $\alpha_{do}$ [‡] |
|---|---|---|---|---|---|
| $Nom_0$ | 0.27 | 0.6 | **2** | **65°** | **45°** |
| $Nom_1$ | 0.09 | 1.66 | 2 | 65° | 45° |
| $Var_0$ | 0.5 | 1.11 | **0.5** | **50°** | **52°** |
| $Var_1$ | 0.5 | 0 | 0.5 | 50° | 52° |
| $Var_2$ | 0.09 | 2.43 | 0.5 | 50° | 52° |

† Dimensionless. ‡ Sexagesimal degrees.

The Nom dimensions' group refers to the parameters used during the model building. $Nom_0$ was the initial model, whereas $Nom_1$ was the model achieved for minimal conicity

and $A_r > 3$ in the second target. The Var dimensions' group is a variant group used to test the accuracy of the model. It changes the axis of $D_f$ and $D_o$, as well as the inlet fluid duct diameter, $D_{df}$. Conicity and eccentricity were swept across a wide set of values, including quasi-non-conicity (Con = 0.09) and non-eccentricity (Ecc = 0).

The advantages and limitations of the CEDG and Solid Edge 2023 approaches were evaluated through the first two targets. In order to simplify the metrics and extension of the third target, we used three relevant points of the intersection curves from flat patterns. Numeric precision will be limited to three decimals (mm).

## 3. Results

*3.1. Surfaces' Intersection and Flattening through Locus-Based Parametric Functions*

3.1.1. Surface-to-Surface Intersections

The encounter or intersection curve between two 3D surfaces, $\mathbf{S_1}$ and $\mathbf{S_2}$, $\mathbf{C}$, may be defined as the set of all 3D points, P, that pertain to both surfaces ($P \in \mathbf{S_1} \cap \mathbf{S_1}$).

We define $\mathbf{S}(\omega)$ as a collection of auxiliary surfaces that intersects both with $\mathbf{S_1}$ and $\mathbf{S_2}$ when the argument $\omega \in \Omega$, where $\Omega$ is a *reference set* on $\Re^n$ for n = 1 or $n = 2$. Accordingly, the point $P(\omega) = \mathbf{C_1}(\omega) \cap \mathbf{C_2}(\omega)$ will pertain to $\mathbf{C}$ for $\mathbf{C_1}(\omega) = \mathbf{S_1} \cap \mathbf{S}(\omega)$ and $\mathbf{C_2}(\omega) = \mathbf{S_2} \cap \mathbf{S}(\omega)$.

If we choose a reference set $\Omega$ such that the collection $\mathbf{S}(\omega \in \Omega)$ sweeps all the points from $\mathbf{S_1}$ and $\mathbf{S_2}$, then we may compute the intersection curve as $\mathbf{C} = \{P(\omega) = \mathbf{C_1}(\omega) \cap \mathbf{C_2}(\omega)$ for all $\omega \in \Omega$ }.

The projection of $P(\omega)$ from $\mathbf{C}$ onto a plane is called $p_\sigma(\omega)$, in such a way that $\sigma$ is the projection curve of $\mathbf{C}$ on that plane. As we may completely define a 3D curve from any two non-degenerated plane projections [18], we may compute the intersection curve $\mathbf{C}$ through their orthogonal projections on the dihedral planes, $\sigma$ (horizontal) and $\sigma'$ (vertical).

Descriptive geometry gives mathematical procedures that solve the $p_\sigma$ projection for any value of $\omega$. The encapsulation (sequence of mathematical entities in a dynamic geometry model) of that series of mathematical procedures through dependent mathematical objects for the parameter $\omega$ delivers a locus entity, which defines the following equation for the horizontal projection curve, $\sigma$:

$$\mathcal{L}_\sigma(\omega) \equiv \sigma = \text{locus}(p_\sigma(\omega), \omega \in \Omega), \tag{2}$$

in which $\mathcal{L}_\sigma$ is a 2D parametric function in $\Omega$, and locus is the mathematical entity that gives the points $p_\sigma$ associated with all values of $\omega \in \Omega$.

The vertical projection, $\sigma'$, is achieved by the same Equation (2) when the horizontal projection of $P(\omega)$, $p_\sigma$, is substituted by the vertical projection, $p'_\sigma$.

To illustrate this method, Figure 3 presents the computation of the intersection between an oblique cone with an oblique cylinder according to Equation (2). The surfaces are defined by their circular sections with a horizontal plane (bases) and their axis (and vertex v' − v in the cone). The vertical contour of the cone has been added to facilitate the interpretation.

We select the collection $\mathbf{S}(\omega)$ of planes that contain the cone vertex, v' − v, and a line parallel to the cylinder axis, called r' − r. As the incidence of this plane onto the horizontal plane is the h' − h point, a plane from this collection may be defined through the s' − s horizontal line, as shown in Figure 3. The line must intersect the two circular bases to ensure that $\mathbf{S}(\omega)$ intersects the two surfaces. The lines h − PLI and h − PLS define the boundary planes of this collection for the relative position of the surfaces. These lines must be tangents to any of the circular bases. In this manner, the PLS point is computed as PLS = If(y(1) < y(3) then point 1 else point 3), where 1 and 3 are presented in Figure 3 and y is the vertical coordinate. A similar condition with points 2 and 4 defines the PLI point. This conditional logic attends any relative position of these oblique surfaces.

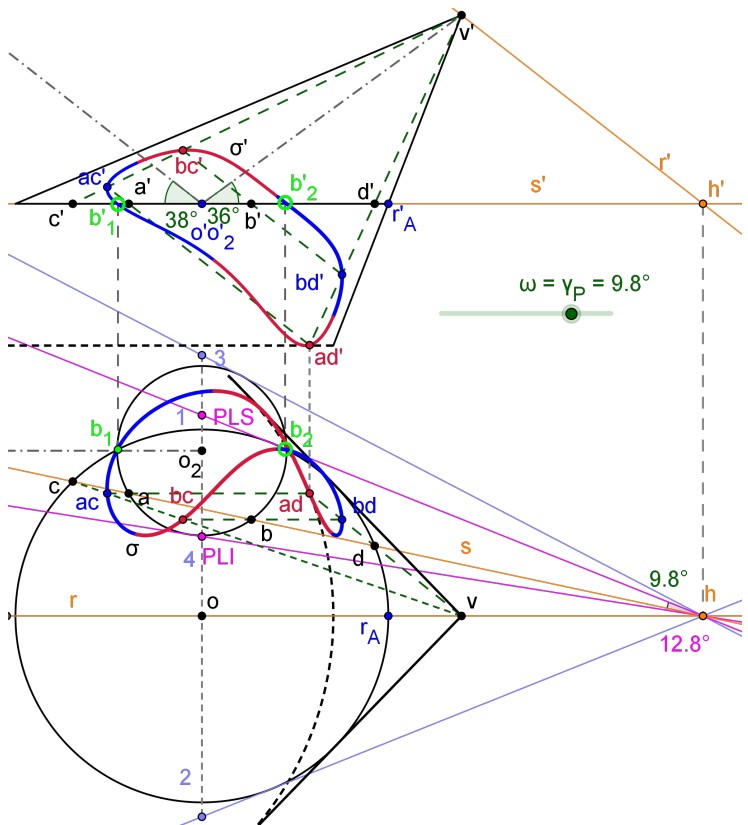

**Figure 3.** Projection curves of cone–cylinder intersection (bite) in a CEDG model.

We have defined $\omega$ as the angle $\gamma_P$ between lines s and h − PLS (9.8° in Figure 3). Therefore, the reference set, $\Omega$, is the interval $[0, \gamma_{Px}]$, where $\gamma_{Px}$ is the angle between lines PLI − h − PLS (12.8° in Figure 3).

A general plane $\mathbf{S}(\omega)$ intersects in two generatrix lines ($\mathbf{C_1}(\omega)$) for the cylinder and other two for the cone ($\mathbf{C_2}(\omega)$), which intersects to give four intersection points ($P_i(\omega) = \mathbf{C_1}(\omega) \cap \mathbf{C_2}(\omega), i = 1, 2, 3, 4$), from $\mathbf{C}$, as shown in Figure 3 (ac′ − ac, bc′ − bc, ad′ − ad, bd′ − bd). Therefore, the vertical and horizontal projections of $\mathbf{C}$, $\sigma'$ and $\sigma$, are given by the following four-leaf parametric curves:

$$\sigma' \equiv \mathcal{L}_{\sigma'}(\omega) = \text{locus}(\{ac', bc', ad', bd'\}, \omega \in \Omega) \tag{3}$$

$$\sigma \equiv \mathcal{L}_{\sigma}(\omega) = \text{locus}(\{ac, bc, ad, bd\}, \omega \in \Omega) \tag{4}$$

The leaves of $\sigma'$ and $\sigma$ have been emphasized through alternating colors (see Figure 3). Figure 3 presents the connection between $\mathbf{C}$ and the cone basis (b′$_1$ − b$_1$ and b′$_2$ − b$_2$), beside the horizontal cone section just below $\mathbf{C}$ (dashed circle), to clarify the solution.

Reducing the distance between the centers of circular bases, o − o$_2$, the bite between surfaces changes to the penetration of the cylinder into the cone, as shown in Figure 4. The boundary planes are now both tangent to the cylinder base (PLS is 3 and PLI is 4). The intersection curve splits into two separate pieces, which are defined by the parametric curves $\mathcal{L}_{\sigma'}(\omega)$ and $\mathcal{L}_{\sigma}(\omega)$ (Equations (3) and (4)) for the vertical and horizontal projections, respectively.

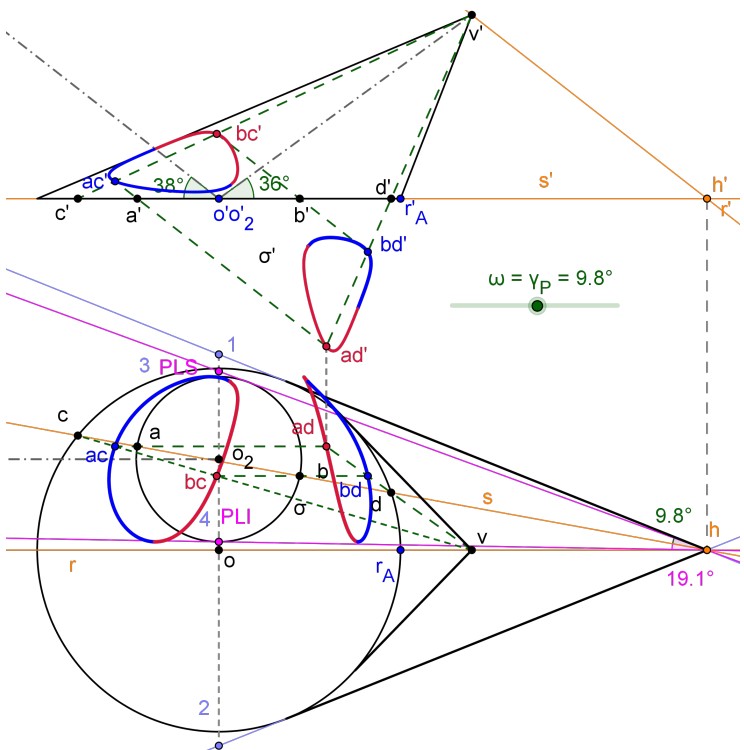

**Figure 4.** Projection curves of the cone–cylinder intersection (penetration) in a CEDG model.

The mathematical-geometric objects described in detail in previous paragraphs and manipulated according DG procedures to calculate the intersections points $P_i(\omega) = C_1(\omega) \cap C_2(\omega), i = 1, 2, 3, 4) = (ac' - ac, bc' - bc, ad' - ad, bd' - bd)$ can be implemented through mathematical entities in the CEDG model. As these points depend on $\omega$, which is a parameter in $\Omega = [0, \gamma_{Px}]$, it is possible to define the loci of Equations (3) and (4) through subsequent mathematical entities that depend on the previous ones. We conclude that this surface-to-surface intersection technique produces a sequence of dependent mathematical entities that may be implemented in a dynamic geometry software in a manner compliant with the CEDG approach, as illustrated in Figures 3 and 4.

An important feature of this technique is that it keeps the geometric integrity of the intersection curve. That is, **C**, defined through the vertical and horizontal projection, $\sigma'$ and $\sigma$, is not an approximation but the true mathematical object, expressed through the parametric functions $\mathcal{L}_{\sigma'}(\omega)$ and $\mathcal{L}_{\sigma}(\omega)$. These functions, in turn, are implicitly coded in the CEDG model, allowing transcendental (non-algebraic) functions.

Although the example is based on quadric surfaces, the method is not restricted to this type of surface. According to descriptive geometry knowledge [18], $S_1$ and $S_2$ may be any type of surfaces for which a collection of auxiliary surfaces $S(\omega)$ exists and produces *computable* intersection curves $C_1(\omega) = S_1 \cap S(\omega)$ and $C_2(\omega) = S_2 \cap S(\omega)$ in $S_1$ and $S_2$, respectively. The term computable is used here to refer to known curves (straight lines, conics, or other ones implemented on well-known mathematical objects), and to curves that may be calculated through the first ones in a recurrent process.

Although a deep analysis of this issue exceeds the scope of this paper, several examples of non-quadric technical surfaces include polyhedral surfaces and even evolution surfaces (e.g., a ship's hull) [18].

### 3.1.2. Surface Flattening

Ruled single curvature surfaces can be flattened without deformation. Here, we present a locus-based technique that gives the exact flat transformation of any 3D curve in this type of surface. It has been implemented in GeoGebra as part of the built-in features of CEDG.

The flat transformation of a curve **C** that pertains to a ruled single curvature surface, **S**, may be formally computed by means of a two-stage procedure. First, a generic generatrix line of **S**, g, that contains a point P ∈ **C** is transformed to the flat domain. This transformation is equivalent to the placement of a line in the flat domain. Secondly, the point P is transformed to the flat domain; that is, it is placed on the transformed g. This two-stage procedure defines a sequence of dependent mathematical objects, which builds the locus entity that defines the flat transform of **C** through a parametric function. Figure 5 shows a scheme of this process for a cylinder and a cone, both oblique with generic directrix.

In the case of a cylinder (Figure 5–left), we must intersect it with a plane perpendicular to the axis, to give the section $\sigma$. The flat transformed of $\sigma$ is the R straight line, perpendicular to transformed generatrix lines. We may use any generatrix line as a reference in the flat domain, as shown in Figure 5 (dashed line). Then, defining $p_\sigma \in \sigma$ as a parameter, it is transformed to $f_R(p_\sigma)$ onto R, ensuring the invariance of lengths of transformed paths. That is, the distance between $f_R(P_\sigma)$ and $f_R(p_{\sigma 0})$ is equal to the length of $\sigma$ between $P_\sigma$ and $P_{\sigma 0}$. The flat transform of the generatrix line through $p_\sigma$ is parallel to the reference one. The function $f_R$ converts its argument from $\sigma$ into the transform in R.

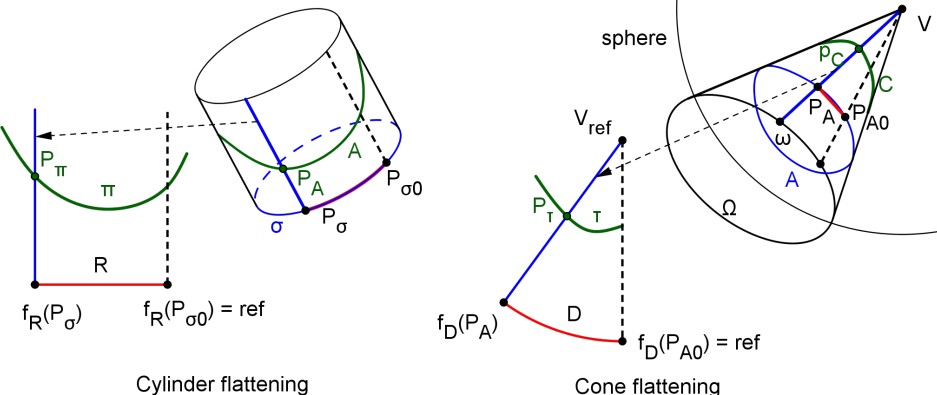

**Figure 5.** Flattening transformation with generatrix line mapping: cylinder (**left**)–cone (**right**).

Finally, the flat transform of $P_A \in \mathcal{A}$ in the cylinder is $P_\pi$, which pertains to the flat transform of its generatrix line, with the distance between $P_\pi$ and $f_R(P_\sigma)$ equal to the distance between $P_A$ and $P_\sigma$.

Considering that descriptive geometry provides mathematical procedures to achieve invariance conditions and manipulate the 3D space through mathematical objects, the flat transform of $\mathcal{A}$, $\pi$, may be computed as:

$$\pi \equiv \mathcal{L}_\pi(\omega) = \text{locus}(P_\pi(f_R(\omega)), \omega \in \Omega),\tag{5}$$

where $\mathcal{L}_\pi$ is a 2D parametric function of $\omega = P_\sigma \in \sigma$, $\Omega$ is a subset of $\sigma$ that defines the cylinder piece to be flattened, and $P_\pi(f_R(\omega))$ is a point of $\pi$ defined by the generatrix line through $f_R(\omega)$.

In the case of an oblique cone as surface **S** to be flattened (Figure 5–right), we first intersect this surface with a sphere centered in the cone's vertex (V) to obtain a warped curve in **S**, $\mathcal{A}$, which is called *support curve*. We know that the flat transform of $\mathcal{A}$ is a circular arch (D) with the radius of the sphere. The flat transform of $P_A \in \mathcal{A}$ is $f_D(P_A) \in D$, with a distance to the reference point $f_D(P_{A0})$ along D equal to the distance from $P_A$ to $P_{A0}$ along $\mathcal{A}$ (see Figure 5). The flat transform of a generatrix line through $P_A \in \mathcal{A}$ contains the flat transform of vertex ($V_{ref}$ placed together with the reference generatrix line) and $f_D(P_A)$.

The flat transform of $P_C \in \mathbf{C}$ in the cone is $P_\tau$, which pertains to the flat transform of its generatrix line, with a distance between $P_\tau$ and $f_D(P_A)$ equal to the distance between $P_C$ and $P_A$. Under the same consideration of Equation (5), the flat transform of **C** ($\tau$) may be computed as

$$\tau \equiv \mathcal{L}_\tau(\omega) = \mathrm{locus}(P_\tau(f_D(\omega)), \omega \in \Omega), \tag{6}$$

where $\mathcal{L}_\tau$ is a 2D parametric function of $\omega \in \Omega$, such that the generatrix line through $\omega$ contains $P_A$, and $\Omega$ is a plane section of the cone that facilitates the mathematical manipulation of this surface.

The variable $\omega \in \Omega$ may be substituted by another variable $t_\omega \in \mathcal{T}$, if there exists any mapping function between $\Omega$ and $\mathcal{T}$.

The computation of the distance from $P_A$ to $P_{A0}$ along $\mathcal{A}$ is not trivial since $\mathcal{A}$ is a warped curve with a geometry that depends on the conical surface. We use the flat transformation of $\mathcal{A}$ as pertaining to a cylinder, taking advantage of the invariance of lengths during the flat transformation.

This method for surface flattening is applied to the conical surface intersected with the cylinder (bite) from the previous CEDG model, between two selected generatrix lines, as presented in Figure 6.

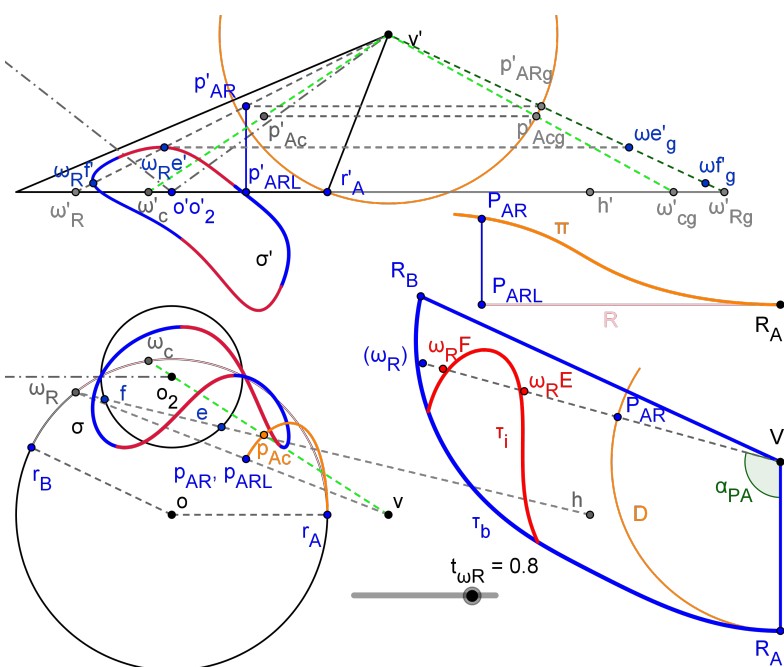

**Figure 6.** Cone surface intersecting with a cylinder (bite, Figure 3) flattened between generatrix lines through $r_A$ and $r_B$. The projections of surfaces include auxiliary lines that help to follow the building process that gives the flat pattern (right–bottom).

Projections are presented in lowercase, whereas flat transformations are in uppercase (except for Greek symbols). In Figure 6, $\omega_R$ is a point in $\Omega$, which is the circular arch from $r_B$ to $r_A$ in the circular base of the conical surface (plane section). It is defined as $\omega_R$ = Point(circular base, $t_{\omega R}$), where $t_{\omega R}$ is a real parameter in $\mathcal{T} = [0, 1]$ and Point is the mapping function between $\Omega$ and $\mathcal{T}$.

The generatrix line through $\omega_R$ intersects the sphere (radius equal to segment $v' - r'_A$) in $p'_{AR} - p_{AR}$. Accordingly, $p_{AR}$ pertains to the horizontal projection of the support curve $\mathcal{A}$, which is calculated through Equation (2) using $\omega_c$ in the circular base, defined between $r_A$ and $\omega_R$. Therefore, the horizontal projection of $\mathcal{A}$ between $r_A$ and $\omega_R$ is locus $(p_{Ac}, \omega_c)$ (orange horizontal projection).

We define a right cylinder from the horizontal projection of $\mathcal{A}$ (bottom base) to the support curve (warped) $\mathcal{A}$. The flat transform of $\mathcal{A}$ as a curve of the cylinder is $\pi$, which is shown in Figure 6 (upper right). According to the Equation (5), $\pi$ is defined as locus($P_{AR}$, $t_{\omega R}$). The distance between $r_A$ and $p_{ARL}$ in the horizontal projection of $\mathcal{A}$ is computed through Perimeter(locus) command.

The distance between $R_A$ and $P_{AR}$ along D (circular arch in cone flat domain, Figure 6 bottom–right) must be equal to the distance between $R_A$ and $P_{AR}$ along $\pi$ curve, which is known. Therefore, $P_{AR}$ in D and its associated generatrix line through ($\omega_R$) may be placed in the flat domain. The flat transformation of the cone base through $R_A$ to $R_B$ may be computed as locus(($\omega_R$), $t_{\omega R}$), according to Equation (6). Finally, the flat transformation of the cone–cylinder intersection inside the flat pattern is defined by two parametric functions, locus($\omega_R F$, $t_{\omega R}$) and locus($\omega_R E$, $t_{\omega R}$), since $\omega_R f'$ − $\omega_R f$ and $\omega_R e'$ − $\omega_R e$ are the intersections of the generatrix line through $\omega_R$ with the cylinder.

The CEDG model updates dynamically to changes in the parameters, which include the modification of the intersection. In this manner, if we reduce the distance between the centers of circular bases $o − o_2$, as was performed in Figure 4, the bite changes to penetration, splitting into two pieces, and the flat pattern dynamically changes to give what shows Figure 7. It must be noted that we have also moved $r_B$ to increase the cone surface that is flattened.

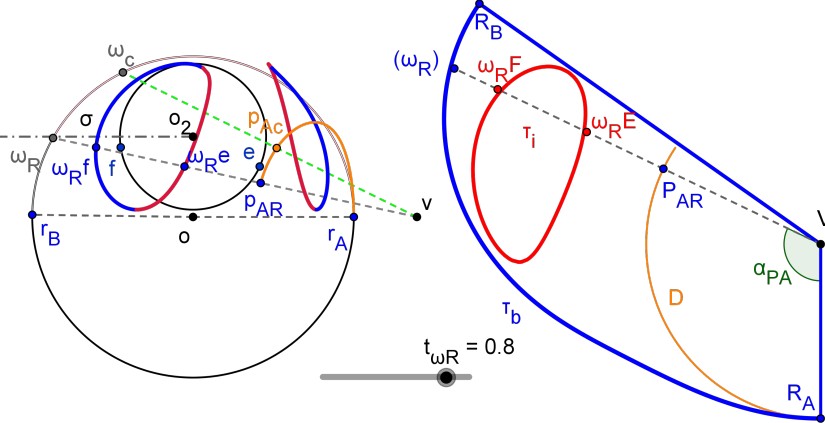

**Figure 7.** Cone surface intersecting with cylinder (penetrated, Figure 4) flattened between generatrix lines through $r_A$ and $r_B$.

We must remark on several issues of this novel surface flattening method. First, the mathematical-geometric procedures that have been described and applied to compute the parametric points ($\omega_R$), $\omega_R F$ and $\omega_R E$ in the flat domain, can be implemented through mathematical entities in CEDG modeling to allow the determination of the loci given by Equation (6). The implementation is performed automatically during the model building process in a dynamic geometry software such as GeoGebra. Second, the technique keeps the geometric integrity of the flattened curves, through the parametric functions: locus(($\omega_R$), $t_{\omega R}$), locus($\omega_R F$, $t_{\omega R}$) and locus($\omega_R E$, $t_{\omega R}$), which in turn are implicitly coded in the CEDG model.

Finally, the technique is limited to ruled single curvature surfaces **S**, in which for any point P in $\mathbf{C} \in \mathbf{S}$ there exists a generatrix line that includes P and may be converted to the flat domain without surface deformation.

### 3.2. Hopper's CEDG Modeling

Figure 8 shows the $D_u − D_f$ connection, calculated according to the technique presented in Section 3.1.1.

We set a collection $\mathbf{S}(\omega)$ of auxiliary planes through the cone vertex (fluid duct) and parallel to the axis of the upper duct (vertical). The argument $\omega$ is a point in the circular base of the upper duct, between $p_3$ and $p_4$ ($\Omega$ is that circular arch). The auxiliary plane from $\mathbf{S}(\omega)$ with $\omega \in \Omega$ intersects the fluid duct in two generatrix lines, $\mathbf{C_1}(\omega)$, and the upper duct in a generatrix line (at the inlet side), $\mathbf{C_2}(\omega)$. These generatrix lines intersect themselves to give the points $P_1$ and $P_2$ through their projections. As seen, the horizontal projection is $p_1 = p_2 = \omega$, whereas the vertical projections are $p'_1$ and $p'_2$.

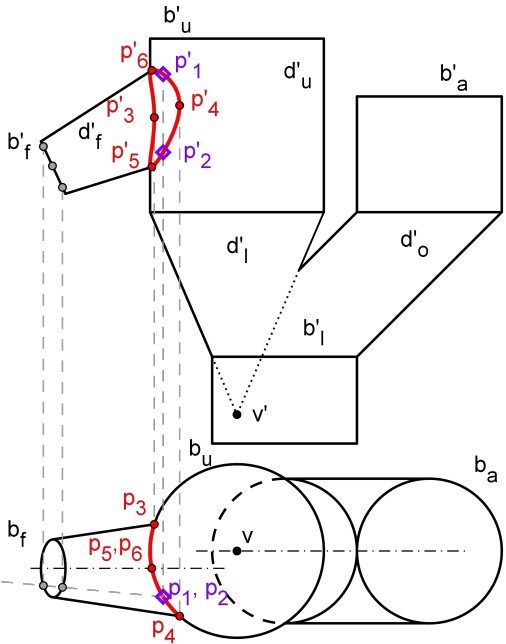

**Figure 8.** CEDG modeling of $D_u - D_f$ connection curve.

Therefore, the $D_u - D_f$ connection curve is given by the parametric functions locus($p'_1$, $\omega$) and locus($p'_2$, $\omega$) (vertical projection, two leaves), and locus($\omega$, $\omega$) = $\Omega$ (horizontal projection, one leaf).

The intersection curve $D_l - D_o$ is presented in Figure 9. It was calculated using a collection of horizontal planes with heights from the low border, $b'_1$, to the height of $Q_5$, as shown. In this manner, we define $\omega = z_h$ (pointed out in Figure 9) and $\Omega = [z(b'_1), z(q'_5)]$. $C_1(\omega)$ and $C_2(\omega)$ are now circles that intersect in $q_1$ and $q_2$ (horizontal projection), and $q'_1 = q'_2$ (vertical projection), and thus the intersection curve is given by the parametric functions locus($q'_1$, $z_h$) = locus($q'_2$, $z_h$) (vertical projection, one leaf), and locus($q_1$, $z_h$) − locus($q_2$, $z_h$) (horizontal projection, two leaves).

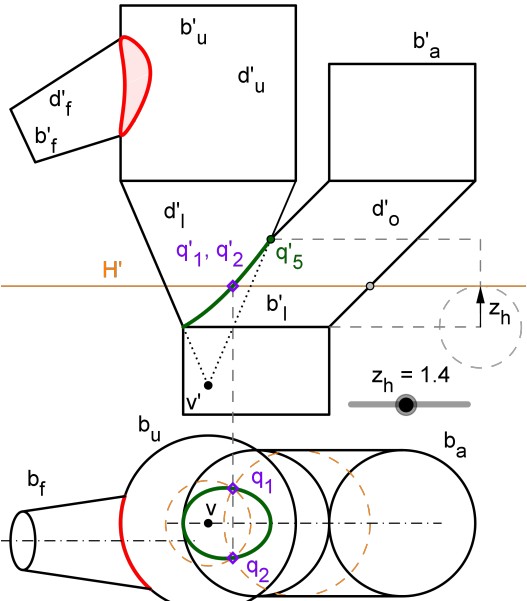

**Figure 9.** CEDG modeling of $D_l - D_o$ connection curve.

The $D_l - D_o$ connection curve is warped despite its horizontal projection seeming elliptic.

Once the connection curves are defined in our CEDG model, the upper and lower ducts may be flattened. The flat pattern of the upper duct was calculated, as indicated in Figure 10. The section perpendicular to the axis of the cylinder is projected as the $b_u$ circle in the horizontal plane. We have taken the generatrix line through A (a' − a) as a reference in the flat domain. We define the Ω subset of the $b_u$ circular section as the circular arch from $p_3$ to $p_4$, which sets the boundaries where the $D_f$ cone intersects with $D_u$ (Figure 10). The generatrix line through the point $\omega \in \Omega$ intersects the $D_u - D_f$ connection in two points, $p'_1$ and $p'_2$, with horizontal projections $p_1 = p_2 = \omega$. Then, they may be transformed to the flat domain thanks to the transformation length invariance to give $P_1$ and $P_2$.

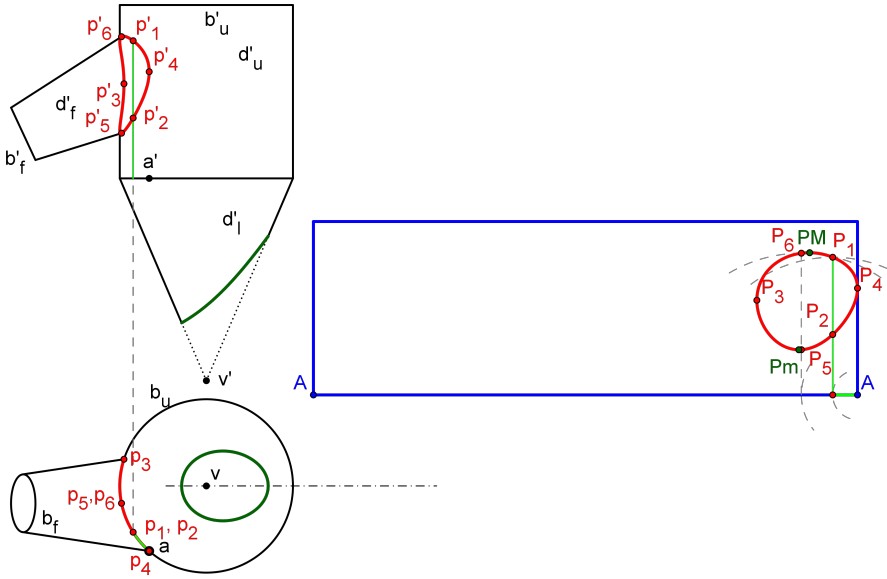

**Figure 10.** Flat pattern of the upper duct ($D_u$) in the CEDG model.

We can apply now the Equation (5) according to the surface flattening technique, which solves the flat transformation of the $D_u - D_f$ connection curve as the parametric functions locus($P_1, \omega$) and locus($P_2, \omega$) with $\omega = p_1 = p_1$ (two leaves).

Figure 10 (right) also shows several relevant points within the intersection curve whose positions with respect to A are used to verify the accuracy of the flat transform. These areas $P_M$ and $P_m$, with maximum and minimum height, $P_3$ and $P_4$, associated with the curve's boundary limits, and $P_6$ and $P_5$ (which are defined below), will be used to compute $A_r$.

The lower duct, $D_l$, is a cone that is flattened according to Equation (6). The process is presented in Figure 11, including the final flat pattern (right). As $D_l$ is a revolution of a circular cone, we have selected the sphere that generates the cone border, $b'_c - b_c$, as support curve $\mathcal{A}$. We use again the point A (a' − a) to set the reference generatrix line, which facilitates the patterns' junction. As $\mathcal{A}$ is a flat curve, it may be also used as a plane section and then $\Omega = \mathcal{A}$. Defining the point q (horizontal projection) from $\Omega$, it may be transported to the flat transformed of $\mathcal{A}$, D, keeping the length between its transform, Q, and A (reference) along D, as shown. The intersection of the generatrix line through q with the $D_l - D_o$ connection curve is $q'_3 - q_3$, which is transported to the flat domain to deliver $Q_3$ (keeping the distance $V - Q_3$). Therefore, the $D_l - D_o$ flat pattern is given by the parametric function locus($Q_3, q$).

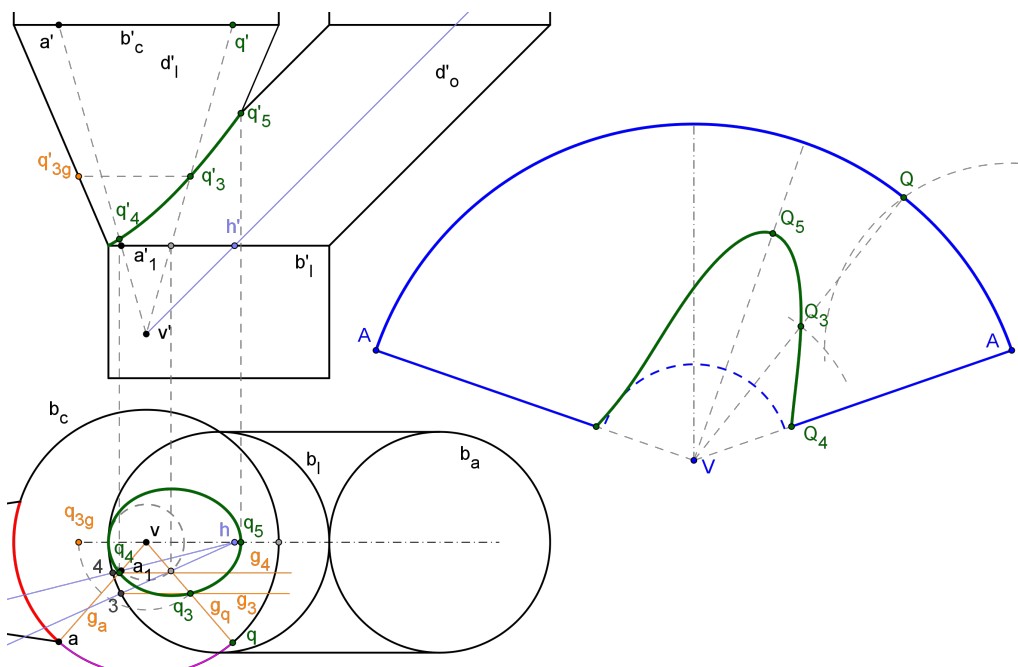

**Figure 11.** Flat pattern of the lower duct ($D_l$) in the CEDG model.

The model solution also shows $Q_4$, which is the intersection of the reference generatrix line with the $D_l - D_o$ connection curve, and its highest point, $Q_5$, which will be used subsequently in the comparative analysis against the CAD model. These points were computed in the hopper projections, using the associated generatrix lines from $D_o$ and $D_l$ and transformed to the flat domain.

Table 2 presents the coordinates of the selected relevant points within the intersection curves' flat patterns for each set of hopper dimensions' values.

**Table 2.** Coordinates of selected points of flat patterns from upper and lower duct (m) and $A_r$ ratio, computed with CEDG for each dimension's groups.

| Dim. Group | $P_{3y}$ | $P_{3z}$ | $P_{4z}$ | $P_{my}$ | $P_{mz}$ | $P_{My}$ | $P_{Mz}$ | $Q_{5g}$ | $Q_{5\alpha}$‡ | $A_r$† |
|---|---|---|---|---|---|---|---|---|---|---|
| $Nom_0$ | 3.487 | 3.275 | 3.691 | 1.979 | 1.563 | 1.633 | 4.921 | 5.440 | 51.728° | 3.259 |
| $Nom_1$ | 3.657 | 3.147 | 4.269 | 2.656 | 2.016 | 1.637 | 4.777 | 5.440 | 39.679° | 3.024 |
| $Var_0$ | 3.478 | 2.276 | 4.010 | 2.554 | 0.972 | 0.977 | 4.268 | 7.464 | 44.246° | 56.361 |
| $Var_1$ | 2.287 | 2.476 | 2.476 | 1.124 | 0.881 | 1.098 | 3.668 | 7.464 | 62.302° | 31.696 |
| $Var_2$ | 2.110 | 2.642 | 4.177 | 1.901 | 2.320 | 0.963 | 4.227 | 7.464 | 38.599° | 15.825 |

† Dimensionless. ‡ Sexagesimal degrees.

The CEDG model of the hopper was visually inspected to propose a surrogate of the discharge area of the fluid duct ($A_o$) that avoids a more complex computation in agreement with the method's requirements (see Section 2). Figure 12a shows the connection curve $D_u - D_f$ after two sequential rotations ($\phi_{ev}$ and $\phi_{eh}$ with the axis through $D_f$ vertex perpendicular to dihedral planes) in a position where the internal segment $P_5 - P_6$ (pertaining to a generatrix from $D_u$ that encounters the $D_f$ contour generatrices in Figure 8) is parallel to the vertical projection plane. It is clear that the second internal segment $P_3 - P_4$ is approximately perpendicular to the first one. The product of the lengths of these segments is the area of a rectangular surface that remains approximately perpendicular to the average streamline. Accordingly, we decided to define $A_o = \overline{P_5 P_6} \cdot \overline{P_3 P_4}$. The last column in Table 2 shows the outlet/inlet area ratio, $A_r$, in the fluid duct.

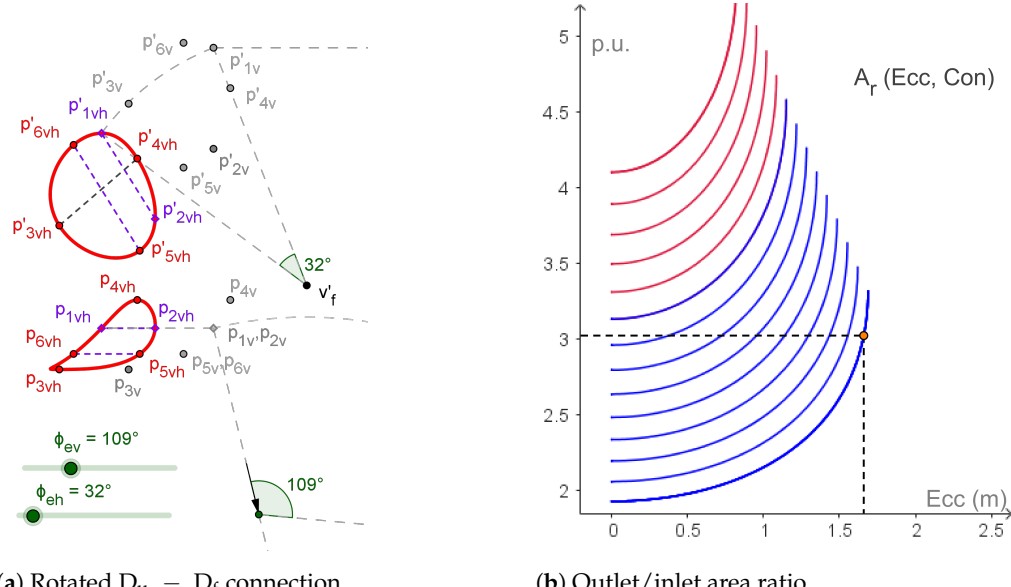

(**a**) Rotated $D_u - D_f$ connection.

(**b**) Outlet/inlet area ratio.

**Figure 12.** Visual inspection of the fluid duct connection (**a**) and numerical relation of its area section, $A_r$, as a function of eccentricity (abscissa) and conicity (0.09–0.25 in blue, 0.27–0.35 in red, increments of 0.02) (**b**) in the CEDG model.

For the values of dimensions used during the model building ($Nom_0$, see Table 1), $\overline{P_3P_4} = 3.320$ m, $\overline{P_5P_6} = 3.339$ m, and thus $A_o = 11.085$ m$^2$. The inlet area is $A_i = \pi D_{df}^2/4 = \pi$ m$^2$, and then $A_r = 3.529$. Although the area ratio is greater than 3 as wished, the conicity could be reduced.

The value $A_r$ as a function of the conicity, Con, and the eccentricity, Ecc, of the fluid duct is computed by the CEDG model and presented in Figure 12b. The conicity values greater than or equal to 0.25 are marked in red. The point marked in that plot achieves a $A_r$ value slightly greater than 3 (3.024) with 1.66 m of eccentricity and a minimum value of conicity (0.09). However, it is possible to even further reduce the tonicity, and the sensitivity of $A_o$ to errors in the hopper's dimensions increases excessively. Therefore, we selected Con = 0.09 and Ecc = 1.66 m as design parameters ($Nom_1$ dimensions' group).

The CEDG model of the hopper associated with the $Nom_1$ design dimensions is presented in Figure 13.

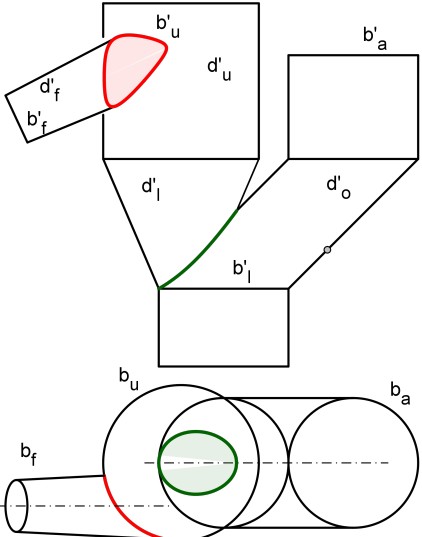

**Figure 13.** Final design of the hopper through CEDG modeling.

### 3.3. Hopper's CAD Modeling

Figure 14 shows the $D_u - D_f$ connection. Both bodies were modeled with the support of the 3D CAD solid modeling software, parametric features, and synchronous technology Solid Edge 2023. We made use of the protrusion command in the solids module to model both the cylinder and the truncated cone. In the case of the latter, a demolding operation was also added to ensure the desired conicity in each case. Once the two solids were created, a thickness of 0.0001 mm was applied because Solid Edge 2023 does not allow one to work with the neutral surface (fiber) of the hopper, as indicated in the methodology pointed out in Prado-Velasco and Ortiz-Marín [17], and the minimum thickness value allowed by this software is the one indicated above.

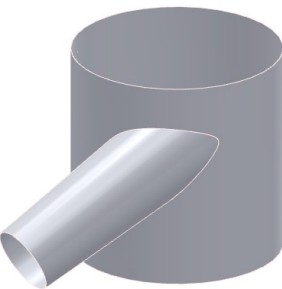

**Figure 14.** Solid Edge modeling of the $D_u - D_f$ ducts system.

In Figure 15, a $D_u$ cylinder is shown together with the intersection curve between it and the previous $D_f$ straight truncated cone. To obtain this part of our whole 3D model in such a way that its flat pattern could be computed, we needed to proceed as follows. First of all, the solid cylinder was created with the protrusion command, and then a thickness of 0.0001 mm was applied. As said before, this is the minimum value allowed by the computer application. Next, a conical cut was applied with the characteristics of the $D_f$ duct, thus obtaining the cylinder together with the intersection curve produced by the truncated cone.

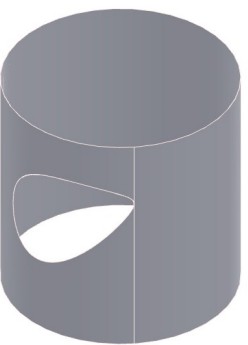

**Figure 15.** Solid Edge modeling of the $D_u$ duct with the $D_f$ connection curve.

Subsequently, it was necessary to make a tiny cut in the lateral surface of the cylinder to generate a slot and be able to open the solid (see Figure 15). This is a fundamental condition if the solid cylinder is required to be converted into a sheet metal module. Once the slot is created, it is possible to access the tools command and select the icon to convert into the sheet metal part. Finally, by selecting the flat pattern button in the tools tab, the flat pattern of the cylinder, together with the conical encounter of the $D_f$ duct, is obtained, as shown in Figure 16, which was obtained by using Solid Edge's draft module. The spatial points indicated are the same as those selected in the CEDG model. Models of Figures 14–16 correspond to the $Nom_1$ design dimensions.

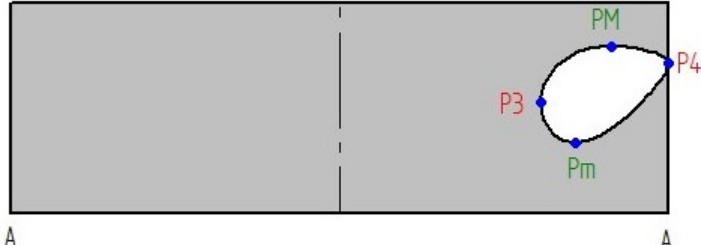

**Figure 16.** Flat pattern of the $D_u$ cylinder together with the intersection curve produced by the $D_f$ truncated cone in the Solid Edge model.

The intersection curve $D_l - D_o$ is presented in Figure 17. As in the case of the preceding cylinder, both bodies were modeled with the Solid Edge 2023 solids module. The truncated straight cone was generated with the protrusion by the revolution command, whereas the oblique cylinder was created by using the protrusion sweep command. Then, a thickness of 0.0001 mm was applied.

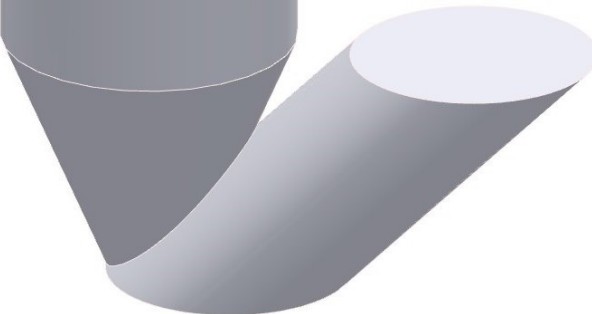

**Figure 17.** Solid Edge modeling of the $D_l - D_o$ ducts system.

The $D_l$ truncated cone was processed in a similar manner to $D_u$ with the aim of solving its flat pattern. The protrusion by the revolution command was used to generate the truncated cone, which was subsequently made a swept hollowing with the same characteristics as the $D_o$ oblique cylinder in order to generate on it the intersection curve produced by this cylinder, as shown in Figure 18. After applying a thickness command of 0.0001 mm, a tiny lateral slot was achieved for the truncated cone in order to open the lateral surface for subsequent application of the sheet metal module. Nevertheless, it was not possible to find a generatrix that would allow that flattening under the bite with the oblique cylinder.

We tried to use the Sheet Metal Body in the Rough command to check if this option could work. After performing several tests, we concluded that the $D_l$ duct can only be flattened when the intersection with $D_o$ is a penetration (two holes) in opposition to a bite (one more complex hole). The Solid Edge surface module was also tested without success because it cannot compute the $D_l$ flat pattern.

Therefore, Table 3 was created with the available information, in which the coordinates of the selected relevant points within the flat patterns of the intersection curves for each set of hopper values are presented together with the values of the outlet (discharge)/inlet area ratio, $A_r$, in the $D_f$ fluid conduit. We used the same surrogate of the discharge area as in the CEDG model to facilitate the comparison.

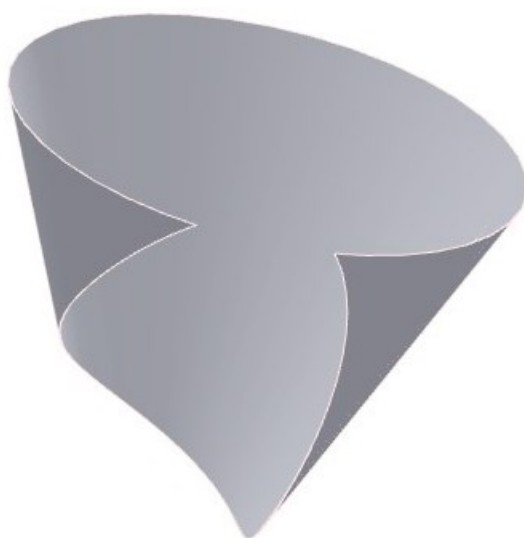

**Figure 18.** Solid Edge modeling of the $D_1$ duct with the connection curve.

**Table 3.** Coordinates of selected points of flat patterns from upper duct (m) and $A_r$ ratio, computed with Solid Edge 2023 for each dimension's groups.

| Dim. Group | $P_{3y}$ | $P_{3z}$ | $P_{4z}$ | $P_{my}$ | $P_{mz}$ | $P_{My}$ | $P_{Mz}$ | $Q_{5g}$ | $Q_{5\alpha}$[‡] | $A_r$[†] |
|---|---|---|---|---|---|---|---|---|---|---|
| $Nom_0$ | 3.487 | 3.275 | 3.691 | 2.022 | 1.563 | 1.675 | 4.921 | - | - | 3.543 |
| $Nom_1$ | 3.658 | 3.147 | 4.270 | 2.666 | 2.015 | 1.632 | 4.777 | - | - | 3.003 |
| $Var_0$ | 3.478 | 2.276 | 4.009 | 2.551 | 0.971 | 0.981 | 4.268 | - | - | 56.618 |
| $Var_1$ | 2.287 | 2.476 | 2.476 | 1.143 | 0.881 | 1.143 | 3.669 | - | - | 31.694 |
| $Var_2$ | 2.111 | 2.642 | 4.177 | 1.831 | 2.312 | 0.761 | 4.228 | - | - | 15.930 |

† Dimensionless. ‡ Sexagesimal degrees.

The obtained values are very similar to those calculated using the CEDG technique, as expected. Notwithstanding, we could not apply the 3D model from Solid Edge to extract the function $A_r$ as a function of the eccentricity and conicity, nor the other parameters.

Finally, the Solid Edge model of the hopper associated with the $Nom_1$ design dimensions is presented in Figure 19.

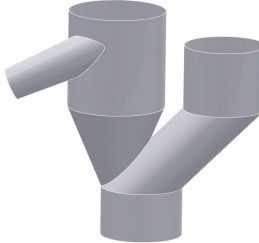

**Figure 19.** Final design of the hopper through Solid Edge modeling.

## 4. Comparative Analysis and Discussion

According to the results obtained, Solid Edge could not complete the hopper modeling. In addition, this CAD tool does not provide an easy method to compute and evaluate $A_r$, as a function of the eccentricity and conicity of our 3D hopper. Each specific target of the second stage of our study is compared as follows.

1.  Feasibility to reach the required models. The 3D model of the hopper that includes the ducts connections was properly obtained both in CEDG and CAD, as shown in Figures 13 and 19. Nonetheless, Solid Edge 2023 was not able to compute the flat

pattern of the lower duct (truncated cone) because this duct encounters the oblique cylindrical duct with an intersection of the bite type. We used different strategies, as described in the Section 3, without success.

2. Once the 3D models were computed, we tried to use them for the analysis of the influence of the geometrical parameters in the outlet/inlet area ratio of the fluid duct, $A_r$, and finally for the optimization of the hopper, to achieve $A_r \geq 3$ in a fast expansion. The CEDG model allowed a visual inspection of the fluid duct—upper duct connection through spatial rotation, as well as the plotting and quantitative computation of the relationship between $A_r$ and any geometrical parameter of the 3D system. We used this feature to plot $A_r$(Ecc, Con) and select the design values Ecc = 1.66 m and Con = 0.09 (Figure 12b). In opposition, we did not find a direct manner to extract the $A_r$ function in Solid Edge 2023.

3. With respect to accuracy, a comparison between Tables 2 and 3 shows that the position of $P_3 - P_4$ (boundary points) in the flat pattern had relative deviations less than 0.01%. In the case of $P_m$ and $P_M$, z relative deviations were lower than 0.02%, whereas y relative deviations were lower than 3.9%. The relative deviations between $A_r$ values were lower than 0.6%, with the exception of the value for $Nom_0$ dimensions' group, which was 8.7%. Values greater than 5% occurred in those cases where a manual selection of some 3D object was needed. We conclude that the accuracy was high in both models.

The failure of Solid Edge 2023 to achieve the flat pattern of a particular surface's configuration agrees with other comparative studies [17]. Although a more complete analysis of this lack exceeds the scope of this paper, the strategy of current CAD technology for surfaces' flattening seems limited to a set of configurations previously defined in the software. This approach is clear in some CAD tools focused on sheet metal, such as LogiTRACE v14.

Our results validate the reliability and accuracy of the new theoretical method presented to compute surfaces' intersections and their flattening for the type of surfaces of the case study. This method takes advantage of the mathematical procedures of descriptive geometry and extends them by means of a locus implicit parametric equation within the novel CEDG approach. The integrity of geometrical objects that characterizes CEDG [11,17] is also kept in this method. Although this study does not evaluate the reliability and accuracy of CEDG and the surface–surface intersection method with non-quadric surfaces, the underlying foundation is not limited to them, as pointed out in the Section 3. The study of the reliability of our method to address the intersections among NURBSs surfaces is a relevant subject that requires more research, since NURBS curves and surfaces are not usually addressed in native form by dynamic geometry software, and thus they must be first implemented as mathematical entities in an efficient computational manner.

The implicit building of parametric functions defining the intersection and flattened curves is an important property shared by other 3D modeling techniques such as the implicit complex (IC) framework of Kartasheva et al. [19] (the third condition of the definition of a graphical cell in the IC). However, CEDG is based on locus-based parametric functions, which differ completely from the IC concept. In addition, CEDG does not have a framework that associates projections with 3D objects. The development of that framework is a current research line in CEDG [20].

CEDG is implemented in a software of dynamic geometry, which provides a different technology from the technology of current CAD systems. Other studies about 3D geometrical modeling have used the locus concept with a different goal. For instance, Gao et al. [21] uses the locus as a mathematical technique to solve some geometrical issues, such as the geometric constraints problem that appears frequently in parametric CAD and other domains, and Oprea and Ruse [22] addressed the computation of some loci problems through descriptive geometry tools.

Other studies have tried to recover the interest in applying the locus concept to CAD software tools. This is the case of the study by Rojas-Sola et al. [23] that improved the set of

available resources to build sketches by incorporating a locus tool into several geometrical cases and implemented them in Adobe Authorware v7 (e-learning authoring tool with a graphical programming language that was officially discontinued in 2007). A more recent study advanced that research line by implementing an algorithm to solve relevant loci cases, which was designed for educational use [24]. Certainly, the development of new mathematical algorithms that improve the locus equation computations is an ongoing research line in industry and research [25].

Although CEDG is a computer parametric 3D approach designed for both educational and professional domains, it is still immature for the professional domain, and it requires more research to improve the usability, computational efficiency, and capability to interact with other CAD tools. Concerning the GeoGebra implementation, the Locus implicit function cannot be used as a general 2D geometric object, and the measurement of lengths along it has limitations [16]. These issues define other ongoing research lines.

**Author Contributions:** Conceptualization, M.P.-V. and L.G.-R.; methodology, M.P.-V.; software, M.P.-V. and L.G.-R.; validation, M.P.-V. and L.G.-R.; investigation, M.P.-V. and L.G.-R.; resources, M.P.-V. and L.G.-R.; writing—original draft preparation, M.P.-V. and L.G.-R.; writing—review and editing, M.P.-V. and L.G.-R.; supervision, M.P.-V. All authors have read and agreed to the published version of the manuscript.

**Funding:** This research received no external funding.

**Data Availability Statement:** No datasets were created, but the CEDG and Solid Edge 2023 models are available on request from the corresponding author.

**Conflicts of Interest:** The authors declare no conflict of interest.

## Abbreviations

The following abbreviations are used in this manuscript:

| | |
|---|---|
| $S_i$ | Data surfaces for $i = 1, 2$ |
| $C$ | Curve in 3D space |
| $\overline{AB}$ | Distance between A and B points |
| A (a′ − a) | Spatial (3D) object (vertical projection—horizontal projection) |
| $P_y$ | horizontal distance between P and right A points in flat pattern (Figure 10) |
| $P_z$ | vertical distance between P and right A points in flat pattern (Figure 10) |
| CEDG | Computer extended Descriptive Geometry |

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
