# Peer review of "Intersection and Flattening of Surfaces in 3D Models through Computer-Extended Descriptive Geometry (CeDG)"

_symmetry, doi:10.3390/sym15050984_

Round 1

Reviewer 1 Report

The paper provides a solution of intersection of two quadric surface: 3D curve, and also its “flattening”, using the CeDG procedure. CeDG is implemented in a software of dynamic geometry (GeoGebra). CeDG uses computer parametric 3D approach, but it differs in a methodology compared to CAD systems.

The procedure showed advantages of CeDG over the CAD capabilities of the Solid Edge 2023 program, which failed to develop the lateral surface of the hopper component, which served as a case study.

Although the paper provides an interesting and useful procedure with evident advantages, my impression is that the key part of the study is overloaded with a large number of details that apply only to the observed case itself, with many explanations that require narrowly profiled knowledge. This does harm to the very principle that is the subject of the study, which is thus relegated to the background. I think it would be much more useful if more attention was paid to the essence of the technique, so it could be more accessible for application in various other cases and professions as well.

The paper is written very carefully and not leaving a single corner uncovered. The introduction is, for example, very nicely written, while in the later text a reader may lose concentration and patience, as a result of the above mentioned.

Remarks:

Term: Surface flattening – why not “development”?

Page 6: Fig 3 and Page 7: Fig 4 - I think it would greatly benefit the clarity of the images if, in the horizontal projections, the contour generatrices of the cones and the contour points where these generatrices are touched by the section curve were entered. (It would be clearer where the boundaries of the surface are, so that a reader would not doubt whether the curve is located on the given cone or has slipped somewhere outside it).

Page 8: Figure 5: it would be clearer if the sphere S were also shown, perhaps with a thinner or fainter line.

The article is well written, although it could have benefited from another check by a native English speaker.

Page 4: Line 140: “The model wants to be optimized” – needs instead of wants?

Page 8: Line 250: “it is placed onto the g transformed” – unclear.

Page 12: line 358: “positions with respect A are used...” – with respect to A?

Author Response

We wish to thank the Reviewers for their constructive criticisms. We have tried to improve the manuscript in response to these. We attach a PDF file with our comments and manuscript modifications.

Reviewer 2 Report

The paper presents an interesting study using Computer Extended Descriptive Geometry (CeDG) for surface intersection and flattening in 3D models. The authors aim to demonstrate the reliability and accuracy of CeDG in solving geometrical problems and present a case study to compare CeDG with CAD (Solid Edge 2023).

The theoretical foundations of CeDG are well-presented, and using locus algebraic equations is an innovative approach to extending the principles of Descriptive Geometry. The authors provide relevant literature references and discuss the potential applications of CeDG in various industries, which adds value to the paper.

However, the introduction could have been more comprehensive in providing background information to understand the research context. Additionally, while the paper presents CeDG as a new approach, it builds upon the foundations of descriptive geometry procedures, which are not entirely new. Therefore, it would have been helpful to discuss the novel aspects of CeDG more explicitly.

The comparison of CeDG and Solid Edge 2023 in the case study is well-conducted and provides interesting results. However, the conclusion may be somewhat exaggerated, as the authors claim that their results validate the reliability and accuracy of the new theoretical method. They only tested the method on a single case study, and it is not clear whether it applies to other geometrical configurations.

Moreover, it would be valuable to investigate the limitations of CeDG, such as its immaturity for the professional domain and the limitations of GeoGebra implementation. The paper briefly acknowledges these limitations, but it would have been beneficial to discuss future research directions in more detail.

In the section discussing the computation of intersection points, more information could be given on how the matrix representations are used to compute the intersection points. 

Overall, the paper presents an interesting approach to solving geometrical problems in 3D modelling. The theoretical foundations of CeDG are well-presented, and the comparison with CAD provides valuable insights. However, the paper would benefit from a more comprehensive discussion of the novel aspects of CeDG, its limitations, and future research directions.

Author Response

(The authors gave the same response as above.)

Reviewer 3 Report

The paper presents the method of finding the intersection of two parametric surfaces using computer-extended descriptive geometry (CeDG). The main idea is to have the intersection curve implicitly defined such that it can be calculated with the required precision later.

While the authors have a substantial amount of work about CeDG, the paper has little information about it, making it harder to understand the method for a reader unfamiliar with the prior work.

If I understand you correctly, you define the surfaces in CeDG explicitly, but the boundary, e.g. intersection curves, can be determined implicitly. It reminds me of the Implicit Complexes framework by Kartasheva et al., you might want to explore that topic as well.

I think the main weakness of the paper is that you do not explain the theory behind the method but instead jump straight into the explanation of it using the example. I understand that the example has been chosen because of practical reasons, but on the other hand, all the intersection curves you are showing as an example resulted in the intersection of algebraic surfaces. The latter problem can be analytically solved as an implicit intersection curve using quadratic equations. If you had two NURBS (note the capitalisation of each letter in this term, you had it wrong in your text) surfaces intersecting, that would be significantly interesting for the general reader. Likewise, the motivation for flattening the surfaces is unclear outside your example. The problem of flattening is very important in computer graphics applications where it is used for texturing. You might want to explore that connection as well.

Having said that, you do good work on explaining all the details of calculating the intersection curve in your particular case, and I have no doubts that it can be easily re-implemented if necessary.

I think the paper is eventually worth being published, but the revision needs to be done to

a)     Properly introduce CeDG,

b)    Explain how your method can be generalised to arbitrary parametric surfaces,

c)     Explain how flattening can also be generalised.

 NURBs -> NURBS

nomenclature <- not sure this term is used in mathematics, consider finding synonime

Author Response

(The authors gave the same response as above.)
